# Artificial Neural Network-Based Abnormal Gait Pattern Classification Using Smart Shoes with a Gyro Sensor

Kimin Jeong [1] and Kyung-Chang Lee [2],*

1   Medical Device Development Center, Daegu-Gyeongbuk Medical Innovation Foundation, Daegu 41061, Korea
2   Department of Intelligent Robot Engineering, Pukyong National University, Busan 48513, Korea
*   Correspondence: gclee@pknu.ac.kr

**Abstract:** Recently, as a wearable-sensor-based approach, a smart insole device has been used to analyze gait patterns. By adding a small low-power sensor and an IoT device to the smart insole, it is possible to monitor human activity, gait pattern, and plantar pressure in real time and evaluate exercise function in an uncontrolled environment. The sensor-embedded smart soles prevent any feeling of heterogeneity, and WiFi technology allows acquisition of data even when the user is not in a laboratory environment. In this study, we designed a sensor data-collection module that uses a miniaturized low-power accelerometer and gyro sensor, and then embedded it in a shoe to collect gait data. The gait data are sent to the gait-pattern classification module via a Wi-Fi network, and the ANN model classifies the gait into gait patterns such as in-toeing gait, normal gait, or out-toeing gait. Finally, the feasibility of our model was confirmed through several experiments.

**Keywords:** gait analysis; artificial neural network; abnormal gait pattern; smart shoes





## 1. Introduction

Changes in gait pattern such as fluctuations in walking speed or swaying of the body are often used as early indicators of cognitive impairment. This is because gait pattern is determined by individual gait characteristics such as lifestyle and differences in skeletal muscles. In addition, walking is an inherent human behavior that constantly changes from infancy to old age, but individual gait patterns are often fixed in childhood if poor habits develop. Therefore, it is necessary to identify an abnormal gait pattern and treat it early. Such findings have increased interest in the design of gait-pattern monitoring and assessment methods [1–6].

Gait-pattern monitoring and assessment systems can be classified into two different types: marker-based and wearable-sensor-based. The marker-based approach obtains walking-motion data using body-attached sensor systems using accelerometer [7,8] or EMG [9], video-based systems [10], active magnetic trackers [11], or optical-marker systems [12]. However, these cannot be used outside the laboratory environment, invade privacy, and are expensive. The wearable-sensor-based approaches [13,14] use accelerometer sensors, pressure sensors, or biosensors worn on clothing or shoes equipped with low-power devices and portable memory for long-term ambulatory monitoring. It is possible to capture and analyze gait information in real time over relatively long distances and outside the laboratory environment [12].

Recently, as a wearable-sensor-based approach, a smart-insole device that has a sensor embedded in the insole has been used to analyze gait patterns [15–18]. Such insoles can be put into any shoe, have the advantage of being compact and inexpensive, and can easily be integrated into small electronic devices. Therefore, smart insoles are currently among the best wearable devices to obtain gait information. In particular, by adding a small low-power sensor and an IoT device to the smart insole, it is possible to monitor the wearer's activity, gait pattern, and plantar pressure in real time, and to evaluate exercise

function in uncontrolled environments. The wearer is not aware of being "tested" because the gait information is obtained during daily life, so any effect of "testing" on his or her walking is minimized, and the data obtained reflect his or her natural gait well.

This paper presents a gait-pattern monitoring device that uses a smart insole, a Wi-Fi network, and an artificial neural network (ANN)-based abnormal gait pattern classification method for use in uncontrolled environments. The sensor-embedded smart soles prevent any feeling of heterogeneity, and Wi-Fi technology allows acquisition of data even when the user is not in a laboratory environment. In this study, we designed a sensor data-collection module that uses a miniaturized low-power accelerometer and gyro sensor, and then embedded it in a shoe to collect gait data. The gait data are sent to the gait-pattern classification module via a Wi-Fi network, and the ANN model classifies the gait into gait patterns such as in-toeing gait, normal gait, or out-toeing gait.

The rest of the paper is organized as follows. Section 2 shows the implementation of the gait-pattern monitoring system using smart shoes with an accelerometer and gyro sensor. A method to classify gait types using an ANN is proposed, and the experimental setup is described in Section 3. Finally, conclusions and plans for future work are presented in Section 4.

## 2. Implementation and Experimental Evaluation of a Gait-Pattern Monitoring System Using Smart Shoes

### 2.1. Characteristics of Toe-Out Angle according to Gait Pattern

Gait is defined as the process of moving the body forward while maintaining the center of the body constant. The movement from the time one heel touches the ground to the time the other heel touches the ground is called a step. The movement from the time one heel leaves the ground to the time it touches the ground again is called a stride. The distance between these points is called the stride length. As the various muscles and joints of both legs act organically and repeatedly, the gait progresses with a balance between the left and right feet [19].

The toe-out angle refers to the angle formed by the foot in the direction in which the body is moving. This angle represents the degree of internal or external rotation of the lower extremity during the stance phase. The toe-out angle is affected by hip-joint movement, the degree of distortion of the tibia and astragalus, and structural abnormalities of the lower extremities. In addition, this angle is used to determine the efficiency and type of walking. When the toes point inward or outward beyond what is normal, the force that pushes the body forward tends to be distributed in the direction in which the toes are pointing, not in the direction of desired movement.

Figure 1 shows the classification of gait patterns by toe-out angle. Here, the x-axis was defined as the direction of walking, the y-axis was defined as the direction toes go out, and the z-axis was defined as the opposite direction to the sole of the shoe. Here, the toe-out angle (yaw angle) was defined as an angle rotated in the y-axis direction of both feet with respect to the z-axis. Normal gait is when the angle of both feet is within the range 0–6.9°; the person's waist is straight, and the two feet are spread slightly outward. In-toeing gait is when the angle of the two feet is within the range of −19.9 to 0°, and the toes are pointed inward. When viewed from the side, the upper body is inclined slightly forward and the lower body slightly upwards. This aggravates the knees and easily fatigues the lower body, causing hip osteoarthritis. Because the insides of the walking shoes are weighted, flat feet and X-legs are often noticed.

Out-toeing gait is when the angle of the two feet is within the range 6.9–31.9°. In other words, it features a step in which the toes extend outwards. An out-toeing gait twists the thigh and calf bones of the leg; the degree of distortion varies depending on the type of movement and the age of the person. If the hip and knee joints are deformed, the outer cartilage is damaged. Severe kyphosis with persistent injuries can shorten and weaken the external thigh muscles [20,21].

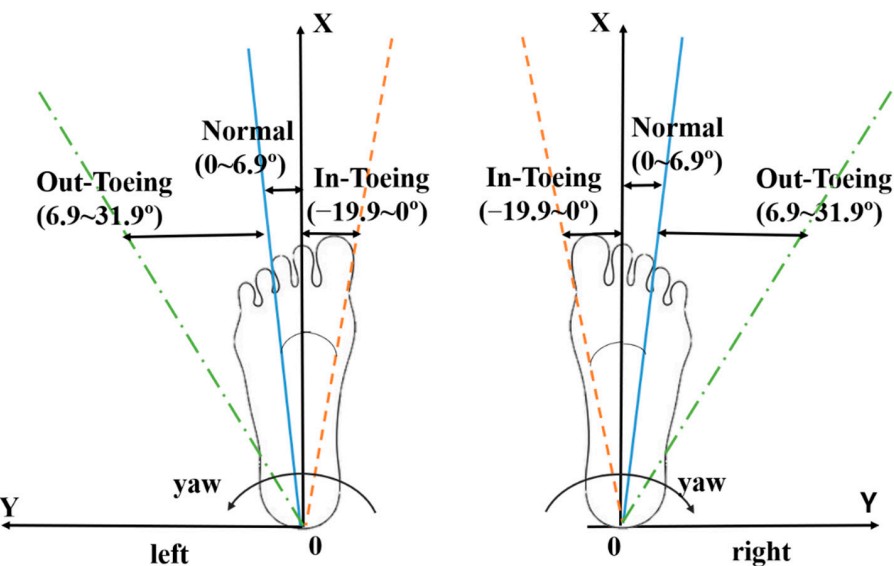

**Figure 1.** Gait pattern according to toe-out angle.

### 2.2. Implementation of Smart Shoes to Measure Toe-Out Angle

Figure 2 shows the smart shoes used to measure toe-out angle. The left side of Figure 2a shows the shoes and the embedded data-collection module. The three-axis acceleration and three-axis gyro sensor data are transmitted over the Wi-Fi network to the gait pattern classification module. The right side of Figure 2a shows the configuration of the gait pattern classification module. The acceleration and gyro sensor data collected by the smart shoes are denoised using a Kalman filter. The angular velocity of the gyro sensor is then integrated with respect to the time axis and estimated as an angle. Thereafter, the angle value is divided into one-step units and input to the ANN classification model, which determines whether a gait pattern is normal or abnormal.

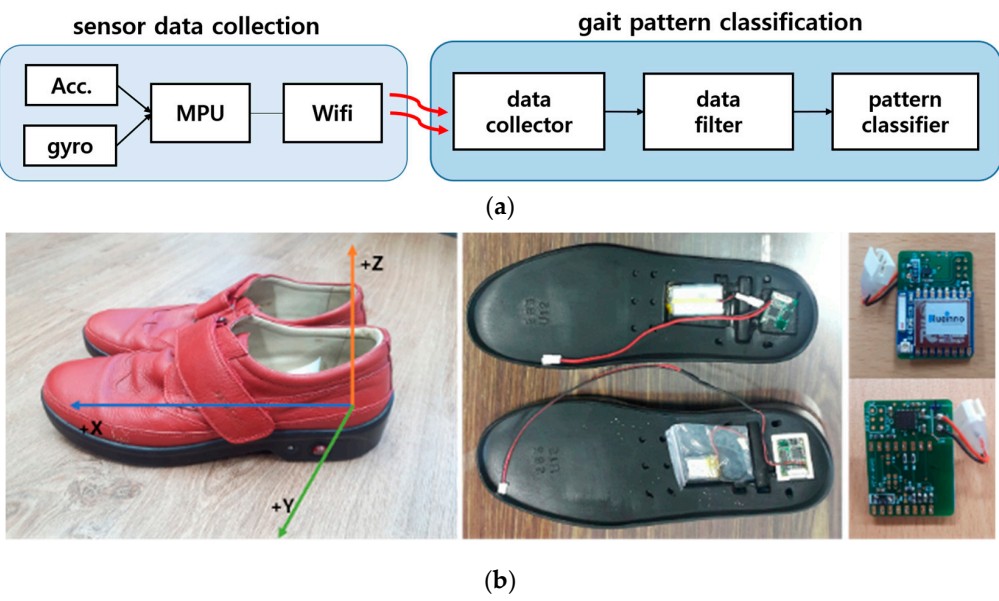

**Figure 2.** Gait pattern classification module. (**a**) Schematic diagram of the overall system. (**b**) The smart shoes with the data-collection module.

Figure 2b shows the data-collection module embedded in a shoe. The module incorporates sensors and batteries, and the shoe is designed to be charged using a dedicated charger. In the figure, the accelerometer is built into the heel of the shoe; the module measures the foot angle based on that between the heel and the second toe.

In this system, a six-axis motion sensor (MPU6050) capable of measuring three-axis acceleration and three-axis gyro sensor values are used. At this time, the x-axis direction of the sensor is the walking direction, the y-axis is the lateral direction, and the z-axis is the height. The sensor values are measured when the wearer is walking on a flat surface, to identify the gait pattern as normal or abnormal.

### 2.3. Experimental Evaluation for Measurement of Toe-Out Angle

The values of the x, y, and z-axis accelerometers and gyro sensors measured by the experimental gait-pattern classification system are shown in Figure 3. To collect sensor values, test subjects were allowed to walk freely for a while before the experiment began and then data measurement commenced after a period of rest. The sensor values were collected over the Wi-Fi network while the subjects were walking straight at normal walking speeds. Via this process, natural walking information can be obtained in uncontrolled environments.

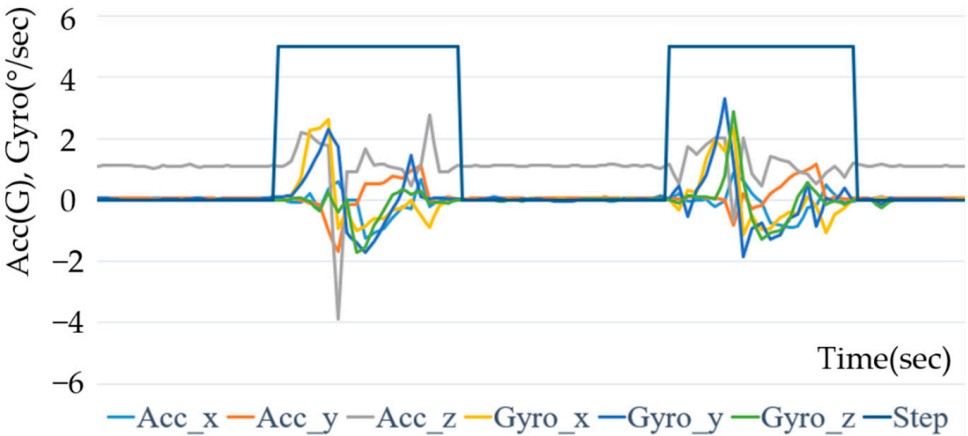

**Figure 3.** Collected raw data of the accelerometer and gyro sensor.

The collected sensor values are truncated step-by-step and automatically stored in the form of step 0, step 1, . . . , step N. When the absolute value of the sum of the signal strength of the three-axis acceleration and three-axis gyro sensors exceeds a predefined threshold of 5000 (output value of 10-bit ADC), it is judged that a new step has begun. Over time, the absolute value of the signal strength falls below the predefined threshold, which is set as the end point of the step. Then, the rotation angular velocities of the x, y, and z axes are integrated and converted into the rotational angles of the x, y, and z axes. Here, the angle of the start point of each step always starts at $0°$ and is provided as an input to the classification module for each one-step unit.

Figure 4 shows the toe-out angles of five steps for an in-toeing gait, a normal gait, and an out-toeing gait. The toe-out angle for each step is slightly different. The test subjects knew that measurements were being made; these values will differ from those of their original gait because they feel they are being measured and then are hesitant to walk. If walking data are obtained only after the test subjects have first been allowed to walk for a sufficient predetermined time, the data will be reliable enough to determine the subjects' normal gait patterns. Figure 4a shows the left foot angle of the x-axis estimated from the gyro-sensor data. In the case of a normal gait, the toe-out angle gradually increases as the foot is lifted from the ground and decreases as it returns to the ground. The maximum toe-out angle of in-toeing, normal, and out-toeing gaits differs slightly, but the overall patterns have a similar shape.

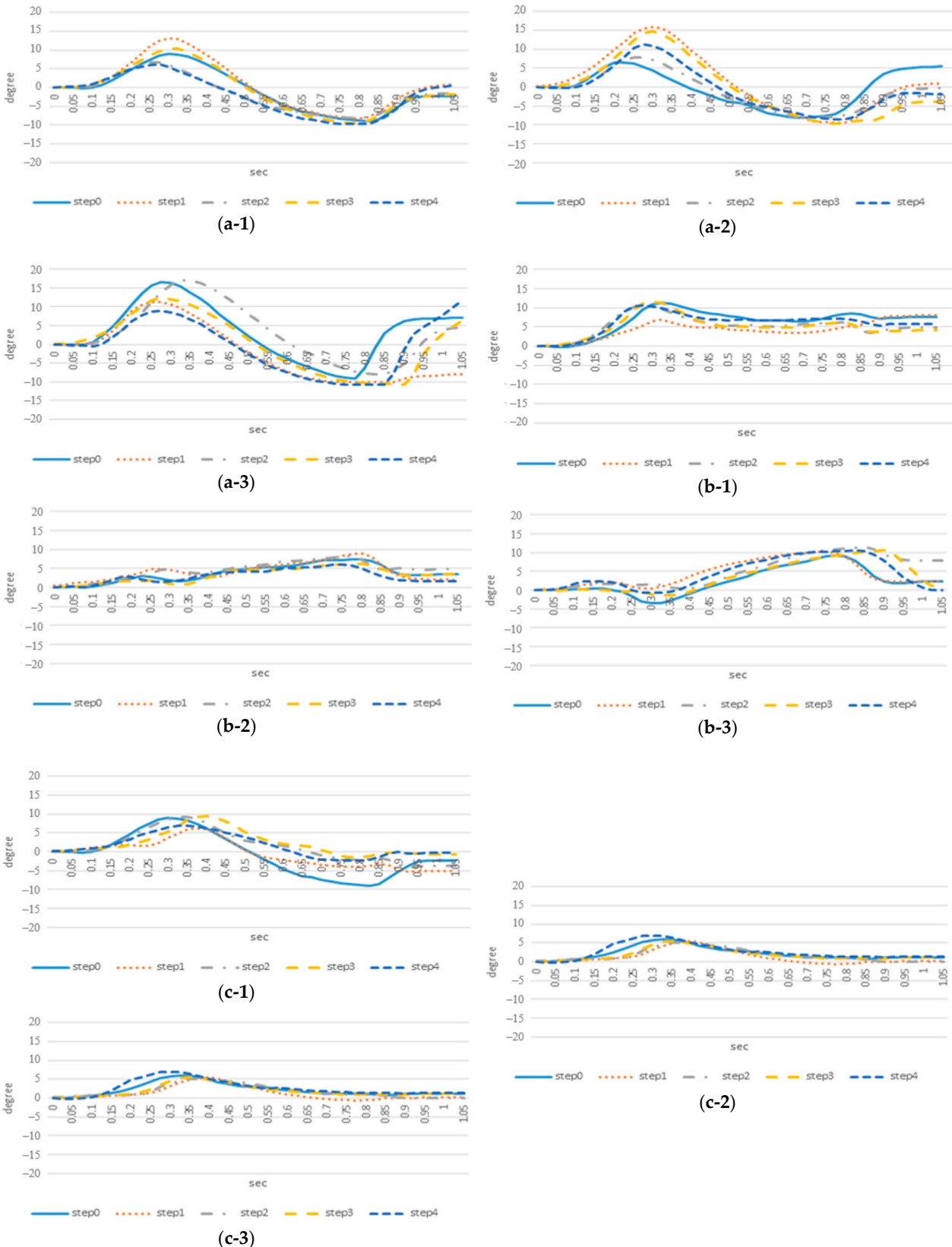

**Figure 4.** Toe-out angles of the left foot; (**a-1**–**a-3**) toe-out angle of x-axis for an in-toeing gait, a normal gait, and an out-toeing gait, respectively, (**b-1**–**b-3**) toe-out angle of y-axis for an in-toeing gait, a normal gait, and an out-toeing gait, respectively, (**c-1**–**c-3**) toe-out angle of z-axis for an in-toeing gait, a normal gait, and an out-toeing gait, respectively.

Figure [4]b shows the angle of the left foot of the y-axis. In the case of a normal gait, the toe-out angle increases initially and then decreases slightly, then gradually increases to the highest point. This pattern occurs because the heel movement does not proceed in a straight line when lifting the foot but turns slightly outward and then goes back inward. In the case of an in-toeing gait, the toe-out angle increases rapidly at the start of walking, reaches a maximum, and then increases again when stepping onto the foot. The reason for this pattern is that the movement of the heel turns from outside to inside while walking. In the case of an out-toeing gait, the toe-out angle decreases initially and then gradually increases; the shape has a peak at the end of the stride. This pattern is because the heel moves from inside to outside. Figure [4]c shows the angle of the z-axis. When the foot is lifted from the ground, the toe-out angle gradually increases, reaches a maximum, and then gradually decreases. As for the x-axis, the maximum value and fluctuation differ depending on the gait pattern, but it can be seen that the three gait patterns have similar tendencies.

The data confirm that the toe-out angle of the y-axis varies slightly with the gait pattern. Based on the changes in the toe-out angle over time, an abnormal gait pattern can be identified. However, as there are differences in the muscle use of each person and as the change in the toe-out angle varies slightly from step to step, a method is needed to determine the gait pattern in various situations.

Figure [5] shows the relationship between the left foot and the right foot in terms of the y-axis toe-out angle. From the figure, it can be seen that the shape of the toe-out angle of the left and right feet is symmetrical with respect to the x-axis. Of course, the maximum value differs because there is a difference in the movements of the right foot and the left heel, but the overall pattern is similar. Based on these characteristics, it is clear that the same method can be used to determine the gait patterns of the left and right feet.

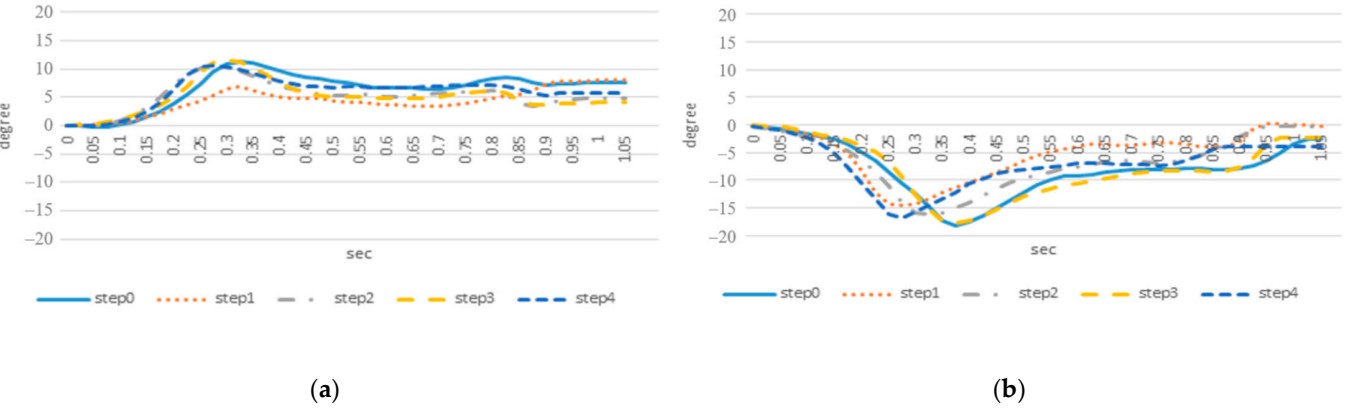

(**a**)                                       (**b**)

**Figure 5.** Toe-out angle of the left foot based on gait patterns. (**a**) Toe-out angle of left foot. (**b**) Toe-out angle of right foot.

## 3. Implementation and Experimental Evaluation of the ANN-Based Gait-Pattern Classification Model

### 3.1. Implementation of Classification Model

In this paper, we propose an ANN classification model that can determine whether a pedestrian's gait pattern is an in-toeing gait, a normal gait, or an out-toeing gait as revealed by acceleration-sensor values measured in smart shoes. Figure [6] shows the abnormal gait pattern classification model using the ANN. In the ANN model, the number of nodes in the input layer is set to 30, the number of nodes in the hidden layer is set to 10, and the number of nodes in the output layer is set to 3.

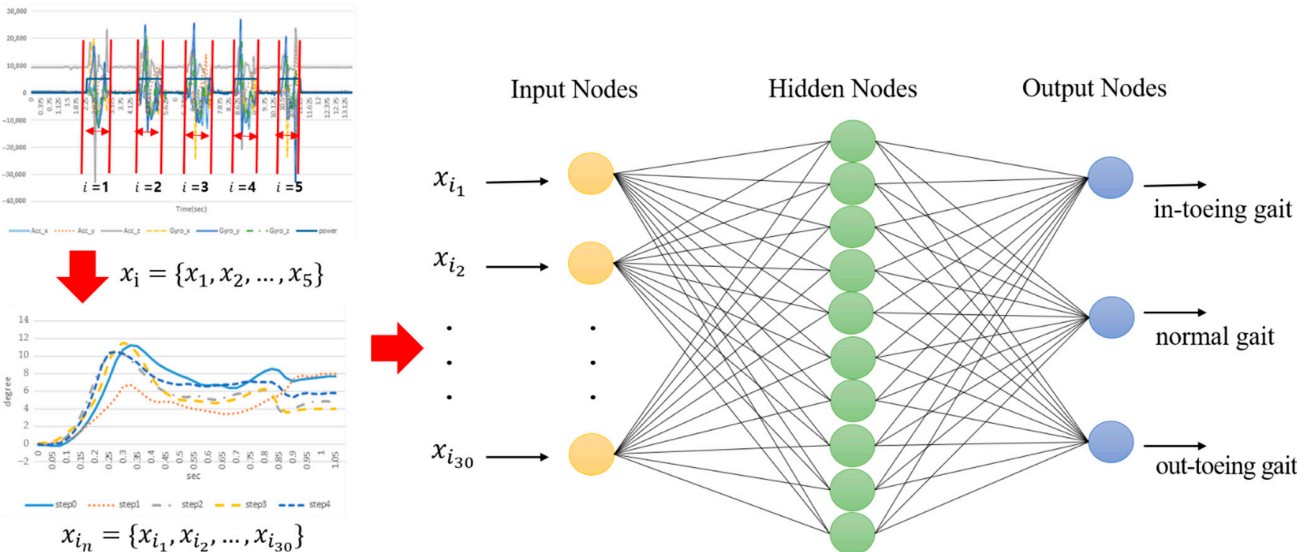

**Figure 6.** The ANN-based abnormal gait pattern classification model.

First, the angular velocity of the y-axis obtained from the smart shoes on the left and right feet is estimated as an angle, and then divided into steps ($x_i$, $i = 1, \ldots, n$). We then acquire 30 angle-data points, the starting point of each step, and store them as a single dataset ($x_{in}$, $n = 1, \ldots, 30$). One angle is acquired per sampling period. The 30-angle dataset (i.e., the dataset for training) is used as the input for each node of the input layer of the ANN model. Since each gait pattern has its own characteristics, the toe-out angle of the y-axis with time can be used to classify gait patterns, using only the angle information acquired from about 30 sampling times initiated by the start of a single step.

Next, we set the number of nodes in the hidden layer to 10. Because angle-data points of 30 tend to change linearly, the number of nodes in the hidden layer is not significantly affected. Accordingly, the number of hidden rays was set to a minimum to minimize the learning time. Finally, the output layer consists of three nodes, each of which assigns an accuracy score of 0 to 1 for all in-toeing, normal, and out-toeing gaits. When the accuracy score of one of the three output nodes is 0.6 or more and the other two are 0.4 or less, that node determines the gait pattern. Gait is judged to be unknown if the scores of two or more output nodes are 0.6 for or no node scores 0.6.

### 3.2. Experimental Evaluation for Gait Pattern Classification

In this study, the experimental environment shown in Figure 7 was constructed to obtain the training dataset. The experiment was conducted with the oral consent of 11 healthy volunteers (two women, nine men, $27.8 \pm 5.1$ years old, and $166.7 \pm 7$ cm tall). Before starting the experiment, all subjects walked freely for two minutes to relax. All subjects wore shoes with built-in data-collection modules and walked 5 m in a straight line to obtain a sampling period of 0.025 s. After recording the subject's gait during the experiment, the complementary angle was observed visually to confirm the gait pattern. Through this process, 55 datasets were obtained for each of five steps of the 11 test subjects.

Figure 8 shows the learning error rate for the ANN model. Back-propagation was used, and the target error rate was set to 0.01. The solid yellow line on the graph is the error rate of the left foot, and the green dotted line is that of the right foot. The target error rate was attained at 78,881 datapoints for the right foot and 95,295 datapoints for the left foot.

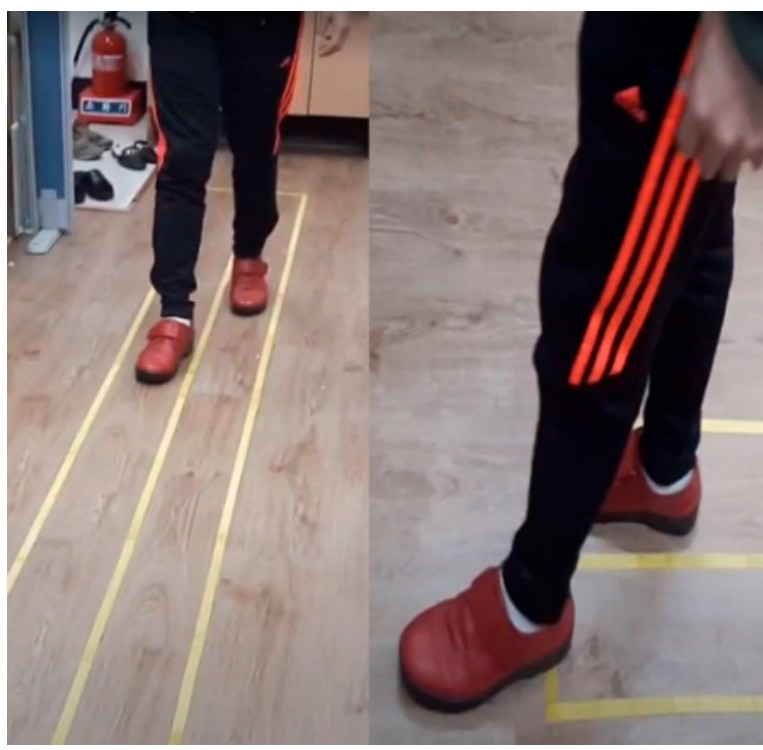

**Figure 7.** Experimental setup for gait measurement using smart shoes.

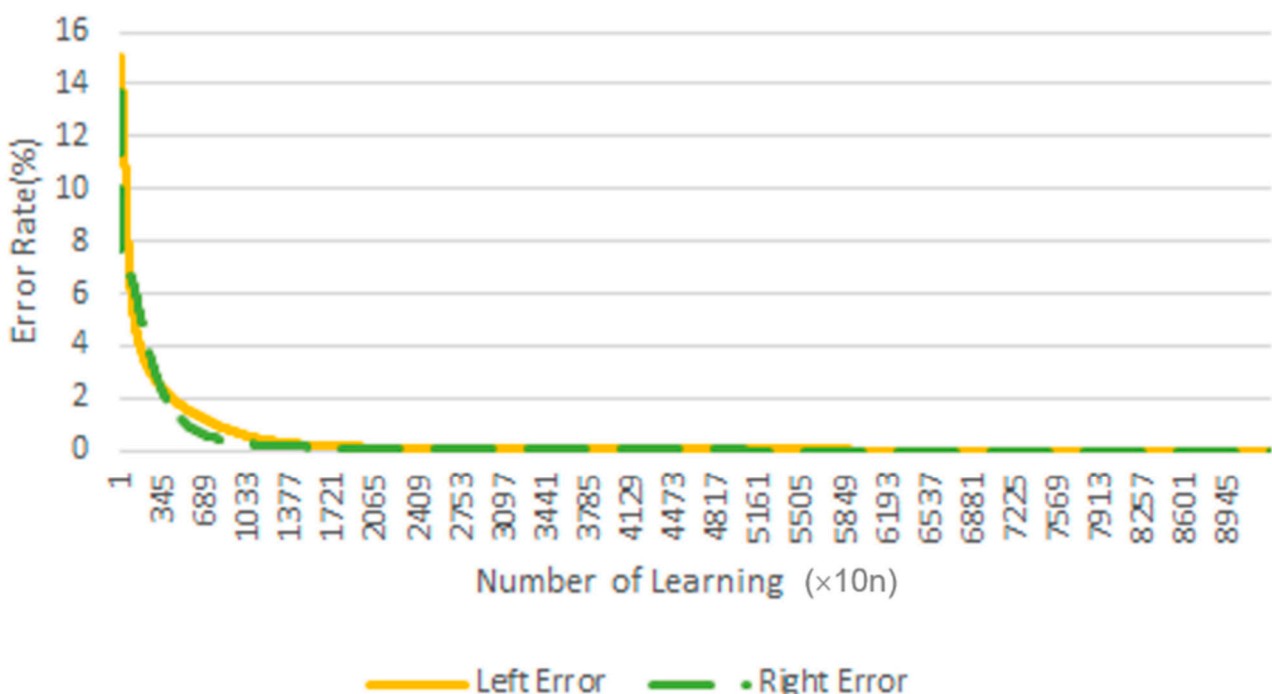

**Figure 8.** Learning error rate of left and right feet.

Finally, to verify the performance of our classification model, the walking data of test subjects who were not included in the training dataset were used. Table 1; Table 2 shows the gait pattern determined by the ANN model. In the table, we verify that the test subject A has out-toeing gait of left foot and normal gait of right foot. Additionally, we know that left and right gait type of the test subject B is in-toeing and normal, respectively.

**Table 1.** Output value of abnormal gait pattern classification model for test subject A.

| | Step of Left Foot | | | | | | Step of Right Foot | | | | | |
|---|---|---|---|---|---|---|---|---|---|---|---|---|
| | 1 | 2 | 3 | 4 | 5 | 6 | 1 | 2 | 3 | 4 | 5 | 6 |
| in-toeing | 0.94 | 5.55 | 2.73 | 0.11 | 2.49 | 2.14 | 0.13 | 0.48 | 0.59 | 14.01 | 11.33 | 29.21 |
| normal | 0.03 | 0.44 | 0.02 | 0.47 | 0.68 | 0.75 | 97.31 | 98.58 | 99.97 | 99.17 | 99.13 | 97.03 |
| out-toeing | 99.39 | 83.81 | 99.15 | 99.67 | 89.72 | 92.98 | 6.84 | 1.31 | 0.03 | 0.02 | 0.03 | 0.03 |

**Table 2.** Output value of abnormal gait pattern classification model for test subject B.

| | Step of Left Foot | | | | | | Step of Right Foot | | | | | |
|---|---|---|---|---|---|---|---|---|---|---|---|---|
| | 1 | 2 | 3 | 4 | 5 | 6 | 1 | 2 | 3 | 4 | 5 | 6 |
| in-toeing | 96.47 | 99.55 | 88.62 | 91.84 | 93.29 | 94.39 | 0.10 | 1.26 | 3.28 | 0.08 | 2.04 | 0.03 |
| normal | 0.01 | 0.36 | 1.06 | 0.02 | 0.02 | 0.01 | 95.95 | 83.78 | 92.47 | 95.16 | 94.25 | 92.47 |
| out-toeing | 0.45 | 0.01 | 0.07 | 1.61 | 1.01 | 1.11 | 11.83 | 4.64 | 3.28 | 15.71 | 1.09 | 3.28 |

In order to validate the performance between classification model's gait pattern and actual gait pattern, we experimentally checked the actual toe-out angle of test subjects. Figure 9 shows the toe-out angle of the feet observed with the naked eye. In the figure, we can verify that results of classification model's and actual gait pattern is same. As a result of verifying the two persons, it can be confirmed that the ANN classification model proposed in this paper is working properly.

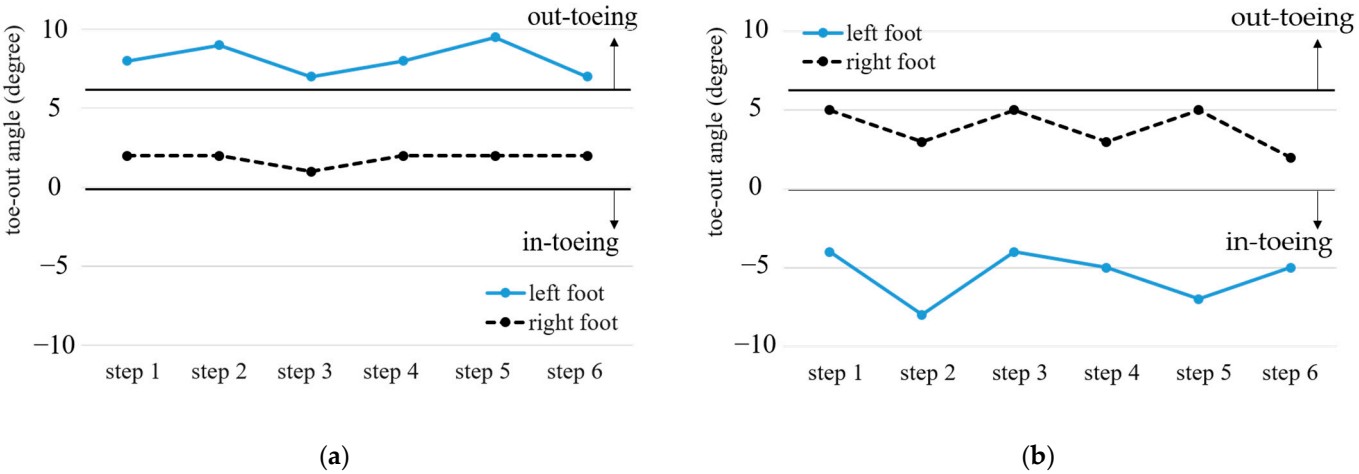

(**a**)                      (**b**)

**Figure 9.** Actual toe-out angle of test subjects. (**a**) actual toe-out angle of test subject A. (**b**) actual toe-out angle of test subject B.

## 4. Conclusions

In this study, we designed a smart shoe with a built-in accelerometer and gyro sensor and proposed a gait-pattern classification system that can be used without the constraints of a controlled environment. Additionally, to classify the gait pattern based on the collected data, the step data were divided into one-step datasets.

As our gait data-collection system embedded in the smart shoes uses low-power Wi-Fi and sensors, it has the advantage of increasing the subject's freedom of walking, unlike other measurement methods that impose restrictions. Additionally, an ANN-based abnormal gait pattern model that can classify gait patterns was designed using this dataset, gait data that are close to those of the normal gait can be acquired. Finally, we find that the feasibility of our model was confirmed through several experiments.

Nevertheless, this research has been limited to evaluating the feasibility of a smart shoes with the classification model of gait pattern. Hence, we intend to develop a model that can classify gait patterns even when the data are collected in various outdoor environments. This would reveal the gait patterns of subjects in outdoor life, and healthy management of gait would be possible. In addition, we intend to develop a system that uses a tactile sensor to measure vertical and horizontal pressures to detect pathological signs in the ankles.

**Author Contributions:** Conceptualization, K.J. and K.-C.L.; methodology, K.J. and K.-C.L.; software, K.J.; validation, K.J. and K.-C.L.; formal analysis, K.J. and K.-C.L.; writing—original draft preparation, K.J.; writing—review and editing, K.-C.L.; supervision, K.-C.L.; project administration, K.-C.L.; funding acquisition, K.-C.L. All authors have read and agreed to the published version of the manuscript.

**Funding:** This work was supported by a Research Grant of Pukyong National University (2021).

**Conflicts of Interest:** The authors declare no conflict of interest.

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
