# Peer review of "Artificial Neural Network-Based Abnormal Gait Pattern Classification Using Smart Shoes with a Gyro Sensor"

_electronics, doi:10.3390/electronics11213614_

Round 1
Reviewer 1 Report
Dear Authors,
I have some comments and suggestions about your paper that should be included on the review version.
1. Line 129. The predefined threshold should be quantified.
2. Line 140-141. The effects of testing should be explained.
3. Line 188-189. Explain why you chose 10 hidden layers.
In Fig1 the angle of all axes X, Y, Z should be defined. In my opinion, should be clear to defined yaw, pitch and roll angle.
Fig4 is unclear, figure caption of each individual figure should be added.
According with Fig.1 the toe-out angle of Y of normal gait should be 90º, why do you use 0º?
Author Response
Thanks for your thoughtful comments.
Please show attached PDF file

Reviewer 2 Report
A major revision is being suggested for the Manuscript ID: electronics-1985127, titled "Artificial neural network-based Abnormal Gait Pattern Classification using Smart Shoes with a Gyro Sensor". The following are the comments that need to be addressed for further consideration:
1. The abstract must be rewritten and must add the major outcomes in numbers.
2. The introduction can be enriched by adding more relevant literature published in the domain, authors also suggested to refer the following articles to understand how present to the research gap and objectives.
a. https://doi.org/10.1016/j.bspc.2021.103429
b. https://doi.org/10.1016/j.engappai.2022.105482
c. https://doi.org/10.1016/j.medntd.2021.100092
d. https://doi.org/10.1016/j.jwpe.2020.101477
e. https://doi.org/10.1016/j.bspc.2021.103346
3. The results must be presented in the heading results and discussion and if the required author can put subheadings. Here under section 3, some results are presented but not discussed. The obtained results must be discussed with the existing recently published literature.
4. What are the parameters used for the assessment of the quality of the developed artificial neural network for this particular classification problem?
5. The conclusion is very generalized and presented as a part of UG project work, it must be rewritten (within 150-200 words) very precise crisp, and clear backed up with the major findings, and can contain 1-2 sentences about future research recommendations.

Author Response

(The authors gave the same response as above.)

Round 2
Reviewer 2 Report
Reviewer’s Comments
The manuscript titled, “Artificial neural network-based Abnormal Gait Pattern Classification using Smart Shoes with a Gyro sensor”, with the Manuscript id: electronics-1985127, is updated as per the provided comments. I recommend that the manuscript be accepted.
I wish the authors great success.
